# X-SNS: Cross-Lingual Transfer Prediction through Sub-Network Similarity

**Taejun Yun[†], Jinhyeon Kim[‡], Deokyeong Kang[†],**
**Seong Hoon Lim[‡], Jihoon Kim[¶], Taeuk Kim[\*†‡]**
[†]Dept. of Computer Science & [‡]Dept. of AI Application, Hanyang University
[¶]Hyundai Motor Company

{tj1616,kimjinhye0n,rkdejrdud88,dmammfl,kimtaeuk}@hanyang.ac.kr,jihoon255@hyundai.com

## Abstract

Cross-lingual transfer (XLT) is an emergent ability of multilingual language models that preserves their performance on a task to a significant extent when evaluated in languages that were not included in the fine-tuning process. While English, due to its widespread usage, is typically regarded as the primary language for model adaption in various tasks, recent studies have revealed that the efficacy of XLT can be amplified by selecting the most appropriate source languages based on specific conditions. In this work, we propose the utilization of sub-network similarity between two languages as a proxy for predicting the compatibility of the languages in the context of XLT. Our approach is model-oriented, better reflecting the inner workings of foundation models. In addition, it requires only a moderate amount of raw text from candidate languages, distinguishing it from the majority of previous methods that rely on external resources. In experiments, we demonstrate that our method is more effective than baselines across diverse tasks. Specifically, it shows proficiency in ranking candidates for zero-shot XLT, achieving an improvement of 4.6% on average in terms of NDCG@3. We also provide extensive analyses that confirm the utility of sub-networks for XLT prediction.

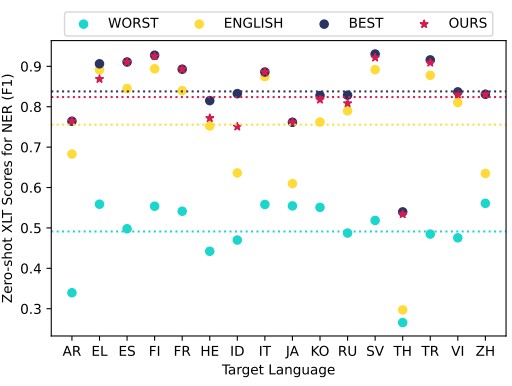

Figure 1: This figure shows that the performance of zero-shot XLT for named entity recognition (NER) varies greatly depending on the language used for task fine-tuning (i.e., the source language). Although English is mostly a default choice for XLT, relying on it often lags behind the best results. In contrast, our method suggests source languages that are nearly optimal. The dotted lines represent average scores for each approach.

## 1  Introduction

One of the aspired objectives in the field of natural language processing (NLP) is to ensure equal treatment of text regardless of the language it originates from (Ruder et al., 2019). This goal is highly desirable in that it promotes the democratization of NLP technologies, making them accessible to everyone worldwide. With the emergence of multilingual language models (Devlin et al. (2019); Conneau et al. (2020); Lin et al. (2022), *inter alia*) that can process over a hundred languages, the ambitious

agenda is partially becoming a reality, although there still remain numerous issues to be addressed.

A prevalent obstacle to the broader use of multilingual models is the need for a considerable amount of task-specific data in the *target language* to properly adapt the models for a particular purpose. This is especially true when following the "pre-training and fine-tuning" paradigm, a general approach for leveraging models of reasonable sizes. That is, merely incorporating support for a specific language in a model is insufficient to address a downstream task in that language; in addition, one needs to secure labeled data in the target language, which is often unfeasible in low-resource scenarios.

Fortunately, the aforementioned problem can be partially mitigated by employing the cross-lingual transfer (XLT) method (Pires et al., 2019; Cao et al., 2020) that exploits supervision from another language dubbed the *source language*. For instance, a multilingual model tuned on English reviews can be directly utilized for classifying the sentiment of

---

[\*]Corresponding author.

reviews in German with decent performance. As seen in the example, English is commonly selected as the source language due to its abundant and diverse data resources available for various tasks.

However, it is not intuitive that English would always be the best option, particularly considering its heterogeneity with non-Latin languages. In fact, recent studies (Lauscher et al., 2020; Turc et al., 2021; Pelloni et al., 2022) have substantiated the conjecture, verifying that certain East Asian languages such as Chinese, Japanese, and Korean exhibit mutual benefits in cross-lingual transfer, surpassing English. In our preliminary study, we also discovered that the performance of zero-shot XLT for NER considerably depends on the choice of the source language (see Figure 1). These findings have naturally led to research into identifying optimal source languages for different configurations of cross-lingual transfer, as well as investigating the factors that contribute to their effectiveness.

Popular strategies for uncovering the optimal source language in cross-lingual transfer include those based on linguistically-inspired features (Littell et al., 2017; Xia et al., 2020), ones that count on statistical characteristics of a corpus of each candidate language (Pelloni et al., 2022), and methods that aggregate various clues for conducting regression (de Vries et al., 2022; Muller et al., 2023). However, the existing techniques have several limitations: (1) they normally require a combination of a myriad of features, each of which is costly to obtain, (2) their effectiveness has only been validated for a limited number of tasks, (3) they present a general recommendation rather than one tailored to individual settings, or (4) they do not reflect the inner workings of the utilized language model.

To address these challenges, this work presents a novel and efficient method called X-SNS, which aims to predict the most appropriate source language for a given configuration in cross-lingual transfer. We introduce the concept of *sub-network similarity* as a *single* key feature, which is computed as the ratio of overlap between two specific regions within a model that are activated during the processing of the source and target language. The main intuition behind the proposed method lies in the notion that the degree of resemblance in the structural changes of model activations, induced by each language, determines the efficacy of XLT.

Our approach is model-based, data-efficient, and versatile. It is closely tied to the actual workings of a base model as it derives the similarity score from the model's internal values, i.e., gradients, instead of relying on the information pre-defined by external resources. In addition, in comparison to previous data-centric methods, our technique is efficient in terms of the quantity of data required. Moreover, we show in experiments that our method is widely applicable across many tasks. It exhibits superiority over competitive baselines in the majority of considered settings. When evaluated in terms of NDCG@3, it achieves an improvement of 4.6% on average compared to its competitors.

The outline of this paper is as follows. We first present related work (§2) and explain the details of our approach (§3). After presenting experimental settings (§4), we perform comprehensive comparisons against baselines (§5) and conduct extensive analyses to provide insights into using sub-network similarity for estimating the performance of XLT (§6). We finally conclude this paper in §7.

## 2 Related Work

### 2.1 Cross-lingual Transfer Prediction

In NLP, the term *cross-lingual transfer* represents a technique of first tuning a model for a task of interest in one language (i.e., *source language*) and then applying the model for input from the same task but the other language (i.e., *target language*). With the introduction of multilingual language models, e.g., mBERT (Devlin et al., 2019) and XLM-RoBERTa (Conneau et al., 2020), and the initial findings suggesting that this technique is viable based on such models (Pires et al., 2019; Wu and Dredze, 2019; Cao et al., 2020), it has emerged as an off-the-shelf solution when only a limited amount of data is available for a specific task in the target language.

There is an ongoing debate in the literature on what factors contribute to the performance of XLT. Karthikeyan et al. (2020) claim that the lexical overlap between languages does not significantly impact its effectiveness, whereas the depth of the employed network is crucial. In contrast, Lauscher et al. (2020) argue that two key aspects influence the level of success in cross-lingual transfer: (1) the linguistic similarity between the source and target languages, and (2) the amount of data used for pre-training in both languages. Similarly, de Vries et al. (2022) exhaustively explore all possible pairs with 65 source and 105 target languages for POS tagging and derive a conclusion that the presence of both source and target languages in pre-training

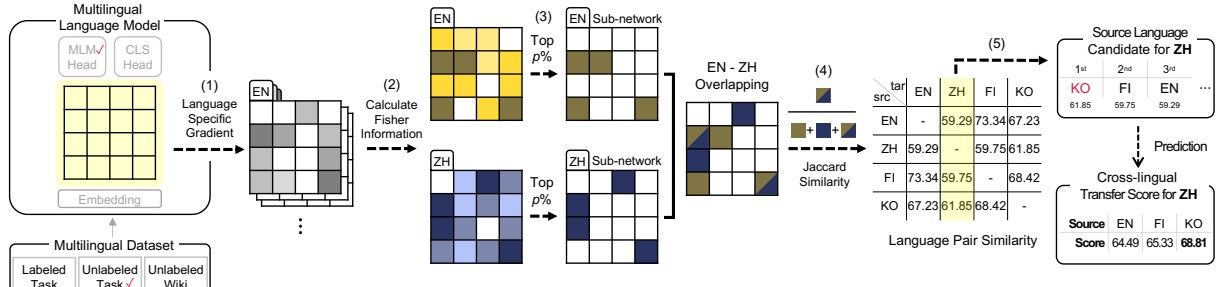

Figure 2: Illustration of the proposed method (**X-SNS**). To extract a language-specific sub-network from a multilingual language model, (1) it first computes the gradients of an objective function with respect to model parameters for each language. Subsequently, (2) it approximates the Fisher Information of the model's parameters using their gradients and (3) selects the top $p\%$ parameters based on the computed values, constructing (binary) sub-networks. Then, (4) the similarity of two languages is defined as the Jaccard coefficient of their respective sub-networks. Finally, (5) the similarity score is used as a proxy for predicting the effectiveness of XLT between two languages. The ✓ symbol denotes the optimal configuration for the proposed method (see the related experiment in §5.1.)

is the most critical factor.

One of the practical challenges in applying cross-lingual transfer arises when multiple source languages are available for the task of interest. In such scenarios, it is necessary to test various variations of a language model by fine-tuning it with data from each source language, which can be cumbersome. To alleviate this burden, several previous studies have proposed methods aimed at estimating the performance of XLT for a given pair of languages and a target task, even without direct fine-tuning of foundation models. Lin et al. (2019) propose an approach to ranking candidate source languages according to a scoring function based on different features including phylogenetic similarity, typological properties, and lexical overlap. Similarly, Muller et al. (2023) estimate numeric transfer scores for diverse tasks by conducting linear regression on linguistic and data-driven features. On the other hand, Pelloni et al. (2022) propose a single feature known as Subword Evenness (SuE), which displays a robust correlation with transfer scores in masked language modeling. The authors claim that the optimal source language for the task is the one in which tokenization exhibits unevenness. In this study, we also present an intuitive method for XLT prediction that is more effective in most cases.

## 2.2 Sub-Network

The notion of sub-networks is broadly embraced in the machine learning community as a means of extracting or refining crucial components of the original network. For instance, Frankle and Carbin (2019) propose the Lottery Ticket Hypothe-

sis, which states that every neural network contains a subset that performs on par with the original network. Similarly, research in the field of model pruning (Han et al. (2015); Liu et al. (2019); *inter alia*) is primarily concerned with iteratively removing a portion of the original network to create its lightweight revision while keeping performance.

Meanwhile, there is another line of research that focuses on modifying only a portion of a neural network while freezing the remaining part. Specifically, Xu et al. (2021) show that better performance and domain generalization can be achieved by tuning only a small subset of a given network. Ansell et al. (2021) propose the separate training of language- and task-specific sub-networks, which are subsequently combined to facilitate the execution of a specific task in the target language.

In this work, we follow the practice established by previous studies regarding sub-networks, but we do not engage in network pruning or directly training a portion of a network. Instead, we identify language-sensitive areas within the network using gradient information and predict the level of cross-lingual transfer based on the overlap of these areas.

## 3 Proposed Method: X-SNS

We present the technical details of our approach, dubbed **X-SNS** (Cross(**X**)-Lingual Transfer Prediction through **S**ub-**N**etwork **S**imilarity). Its primary objective is to suggest proper source language for XLT, eliminating the need for developing separate copies of language models fine-tuned by task data in each language. To this end, we propose utilizing the similarity between a pair of language-specific

sub-networks as an indicator to estimate the compatibility of two languages in XLT. The overall procedure of the method is illustrated in Figure 2.

The core mechanism of X-SNS lies in the derivation of sub-networks customized for individual languages and the computation of similarity between a pair of sub-networks. As a crucial component in the construction of our targeted sub-network, we introduce the Fisher information (Fisher, 1922), which provides a means of quantifying the amount of information contained in parameters within a neural network (Tu et al., 2016; Achille et al., 2019). Concretely, we derive the (empirical) Fisher information of a language model's parameters as follows.[1] First, given a data point $(\mathbf{x}, y)$ from the true data distribution—input $\mathbf{x}$ and output $y$—we define the Fisher Information Matrix $F$ for parameters $\boldsymbol{\theta}$ as:

$$F(\boldsymbol{\theta}) = \mathbb{E}\left[\left(\frac{\partial \log p(y|\mathbf{x}; \boldsymbol{\theta})}{\partial \boldsymbol{\theta}}\right)\left(\frac{\partial \log p(y|\mathbf{x}; \boldsymbol{\theta})}{\partial \boldsymbol{\theta}}\right)^{\top}\right].$$

Second, since it is practically intractable to derive the exact $F$, we assume this is a diagonal matrix, aligning with the approaches employed in the related literature (Sung et al., 2021; Xu et al., 2021).[2] As a result, we obtain the Fisher information vector $\mathbf{f} = \mathrm{diag}(F(\boldsymbol{\theta}))$. Finally, we approximate $\mathbf{f}$ with $\tilde{\mathbf{f}}$, which is the average of Monte Carlo estimates of $\mathbf{f}$ based on a given corpus $D = \{(\mathbf{x}_j, y_j)\}_{j=1}^{|D|}$. Formally, the (approximated) Fisher information of the $i^{\text{th}}$ parameter in $\boldsymbol{\theta}$ is formulated as:

$$\tilde{\mathbf{f}}^{(i)}(\boldsymbol{\theta}) = \frac{1}{|D|} \sum_{j=1}^{|D|} \left(\frac{\partial \log p(y_j|\mathbf{x}_j; \boldsymbol{\theta})}{\partial \boldsymbol{\theta}^{(i)}}\right)^2.$$

We utilize the final vector $\tilde{\mathbf{f}}$, which has the same dimensionality as $\boldsymbol{\theta}$, as a source of knowledge to assess the importance of the corresponding model parameters. Each element of this vector can also be interpreted as the square of the gradient with respect to a model parameter, positioning our method within the realm of model-oriented approaches.

As the next step, we transform $\tilde{\mathbf{f}}$ into a binary vector that represents the structure of the sub-network we intend to extract. Specifically, each component of the sub-network vector $\mathbf{s}$ given $\tilde{\mathbf{f}}$, say $\mathbf{s}^{(i)}$, is specified as 1 if $\tilde{\mathbf{f}}^{(i)}(\boldsymbol{\theta})$ is in the top $p\%$ of all element values in $\tilde{\mathbf{f}}$, 0 otherwise. In other words, we identify a group of parameters within the network whose importance scores are positioned in the top $p^{\text{th}}$ percentile. Note that the proposed method only considers the relative significance of each parameter in terms of its Fisher information, ignoring its actual values. By doing so, X-SNS not only avoids the need for extra training of base models but also ensures its efficiency. This also sets it apart from previous work that employs a similar framework for model pruning or task fine-tuning.

There remain two unspecified factors that play a vital role in acquiring language sub-networks using X-SNS. The first is the choice of the corpus $D$ used for obtaining $\tilde{\mathbf{f}}$. We utilize a corpus from a language of our interest as the basis for deriving a sub-network specific to that language. Besides, we test two types of corpora for each language: a task-oriented corpus ($\mathcal{T}$) and a general domain corpus ($\mathcal{W}$), i.e., Wikipedia. As shown in §5, a task-oriented corpus guarantees superior performance, while a general one enables a broader application of the approach with satisfactory results. In experiments, we also confirm that our method is data-efficient, being nearly optimal when $|D| \leq 1000$.

The second factor is the output distribution of the model $p(y|\mathbf{x}; \boldsymbol{\theta})$. In previous studies (Xu et al., 2021; Foroutan et al., 2022), $p(y|\mathbf{x}; \boldsymbol{\theta})$ is usually defined as a task-specific layer ($\mathcal{T}$), e.g., a classifier, implemented on the base model, requiring additional training of the top module. However, as we prioritize the efficiency of the algorithm, the extra fine-tuning is not considered in our case, even in the environment where a labeled, task-oriented corpus is available. Instead, we imitate such an avenue with a random classifier combined with the cross-entropy loss for the target task. Meanwhile, another realistic direction of defining $p(y|\mathbf{x}; \boldsymbol{\theta})$ is to rely on the pre-training objective of the base model, i.e., (masked) language modeling ($\mathcal{L}$; Ansell et al. (2021)). This not only eliminates the need for task supervision but also allows one to recycle the language modeling head built during the pre-training phase. We thus regard $\mathcal{L}$ as the primary option.[3] Refer to §5.1 where we conduct a related analysis.

Lastly, given a pair of sub-networks $\mathbf{s}_s$ and $\mathbf{s}_t$ for the source ($s$) and target ($t$) languages respectively, the similarity of the languages is calculated as the Jaccard similarity coefficient between the two vectors: $|\mathbf{s}_s \cap \mathbf{s}_t|/|\mathbf{s}_s \cup \mathbf{s}_t|$. In practice, a set

---

[1]As depicted in Figure 2, the model parameters $\boldsymbol{\theta}$ do not include the embedding and task-specific layers in our setting.
[2]For more technical details on the approximation process, we refer readers to Kirkpatrick et al. (2017).

[3]For masked language modeling, we set the mask perturbation ratio as 0.15, making it identical as in pre-training.

of resource-rich languages, as well as the target language for which we aim to perform XLT, are provided. We then proceed by computing the similarity of the target language and each candidate language in the set, and sort the candidates based on the similarity scores in descending order.

## 4 Experimental Settings

### 4.1 Tasks and Datasets

To assess the overall effectiveness of the proposed method across configurations, we employ five tasks from the XTREME benchmark (Hu et al., 2020): named entity recognition (NER), part-of-speech (POS) tagging, natural language inference (NLI), paraphrase identification (PI), and question answering (QA). We explain the details of each task and the corresponding dataset in Appendix A.

### 4.2 Baselines

We consider three distinct categories of methods as baselines: linguistic (L2V), statistical (LEX, SuE), and model-based (EMB). We refer readers to Appendix B for more details on each approach.

**Lang2Vec** (**L2V**; Littell et al. (2017)) provides a diverse range of language representations, which are determined based on external linguistic databases.[4] We utilize the "typological" vectors within the suite, encompassing syntax, phonology, and inventory features. The cosine similarity of two language vectors is regarded as an indicator that predicts the efficacy of XLT between the respective languages. On the other hand, **Lexical Divergence** (**LEX**) characterizes the lexical similarity between two languages. Inspired by Muller et al. (2023), we specify this as the Jensen-Shannon Divergence between the subword uni-gram frequency distributions of the source and target languages. Meanwhile, Pelloni et al. (2022) propose the **Subword Evenness (SuE)** score that evaluates the evenness of word tokenization into subwords. The authors claim that the language with the lowest SuE score (i.e. one showing the most unevenness) is the best candidate for XLT. We compare their suggestion with that of our method. Finally, Muller et al. (2023) propose employing the cosine similarity between two language vectors. In contrast to L2V, these vectors are obtained by taking the average of sentence embeddings computed by language models. We call this **Embedding Similarity (EMB)**.

[4]e.g., The WALS database (https://wals.info/).

| Task | $D$ | $p(y\|\mathbf{x}; \boldsymbol{\theta})$ | Pearson | Spearman | Top 1 | NDCG@3 |
|------|-----|------|---------|----------|-------|--------|
| **NER** | $\mathcal{T}$ | $\mathcal{T}$ | 70.67 | 52.94 | 33.33 | 77.42 |
|  | $\mathcal{T}$ | $\mathcal{L}$ | **78.80** | **58.20** | **39.22** | **78.12** |
|  | $\mathcal{W}$ | $\mathcal{L}$ |  |  |  |  |
| **POS** | $\mathcal{T}$ | $\mathcal{T}$ | 65.76 | 51.42 | 29.44 | 80.01 |
|  | $\mathcal{T}$ | $\mathcal{L}$ | 72.55 | **65.52** | **33.33** | 83.73 |
|  | $\mathcal{W}$ | $\mathcal{L}$ | **74.20** | 62.65 | 32.78 | **84.43** |
| **NLI** | $\mathcal{T}$ | $\mathcal{T}$ | 16.97 | 20.96 | 9.63 | 67.49 |
|  | $\mathcal{T}$ | $\mathcal{L}$ | **24.47** | **31.52** | **11.11** | **68.73** |
|  | $\mathcal{W}$ | $\mathcal{L}$ | 9.12 | 3.12 | 3.70 | 58.30 |
| **PI** | $\mathcal{T}$ | $\mathcal{T}$ | 18.41 | 16.16 | 19.05 | 73.02 |
|  | $\mathcal{T}$ | $\mathcal{L}$ | **73.59** | **64.47** | **52.38** | **89.82** |
|  | $\mathcal{W}$ | $\mathcal{L}$ | 41.14 | 45.01 | 25.40 | 83.49 |
| **QA** | $\mathcal{T}$ | $\mathcal{T}$ | 26.86 | 23.86 | 33.33 | 75.80 |
|  | $\mathcal{T}$ | $\mathcal{L}$ | **72.58** | **68.15** | **50.00** | **87.95** |
|  | $\mathcal{W}$ | $\mathcal{L}$ | 49.07 | 45.49 | 9.72 | 81.51 |

Table 1: According to the selection for the corpus $D$ and the output distribution $p(y|\mathbf{x}; \boldsymbol{\theta})$, the proposed method suggests three variations of sub-networks. For $D$, there exist two options—$\mathcal{T}$: task-specific data and $\mathcal{W}$: text from Wikipedia. Meanwhile, $p(y|\mathbf{x}; \boldsymbol{\theta})$ also have two choices—$\mathcal{T}$: output from task-oriented layers, which requires labeled data, and $\mathcal{L}$: the distribution from (masked) language modeling. The combination of $D=\mathcal{T}$ and $p(y|\mathbf{x}; \boldsymbol{\theta})=\mathcal{L}$ generally produces the best outcomes.

### 4.3 Configurations and Metrics

We employ XLM-RoBERTa_{Base} (Conneau et al., 2020) as our base language model. As the proposed method operates on a small set of data instances, we randomly sample a subset from an available corpus. In order to mitigate potential bias arising from data selection, we execute the method three times with different random seeds and consider its averaged outcomes as its final result. Meanwhile, it is also necessary to compute gold-standard XLT scores, which serve as the standard for assessing the power of prediction methods. To this end, for a given pair of source and target languages and the specific task at hand, we fine-tune the model three times using the source language data. We then evaluate the fine-tuned ones on the target language and calculate the average value, thereby alleviating the instability inherent in neural network training. Note that we focus on the zero-shot XLT setting.

Four metrics are introduced for evaluation. The Pearson and Spearman correlation coefficients are used to gauge the relationship between the gold-standard scores and the predicted scores suggested by each method. We also utilize the Top 1 accuracy to evaluate the ability of each method in identifying the best source language for a given target language. Lastly, Normalized Discounted Cumulative Gain (NDCG; Järvelin and Kekäläinen (2002)) is leveraged to test the utility of the ranked list of

| Task (Dataset) | # | Method | Pearson | Spearman | Top 1 | NDCG@3 |
|---|---|---|---|---|---|---|
| **NER** (WikiANN) | 17 | Lang2Vec (L2V; Littell et al. (2017)) | 14.58 | 17.73 | 13.73 | 62.35 |
| | | Lexical Divergence (LEX; Muller et al. (2023)) | 67.55 | 53.92 | 23.53 | 76.20 |
| | | Subword Evenness (SuE; Pelloni et al. (2022)) | 30.33 | 10.29 | 3.92 | 47.46 |
| | | Embedding Similarity (EMB; Muller et al. (2023)) | 78.18 | 49.52 | 31.37 | 76.06 |
| | | **Ours (X-SNS)** | **78.80** | **58.20** | **39.22** | **78.12** |
| **POS** (UD) | 20 | Lang2Vec (L2V; Littell et al. (2017)) | 67.62 | 56.97 | 23.33 | 78.06 |
| | | Lexical Divergence (LEX; Muller et al. (2023)) | 48.29 | 39.78 | 28.33 | 77.44 |
| | | Subword Evenness (SuE; Pelloni et al. (2022)) | 42.83 | 48.99 | 0.00 | 33.26 |
| | | Embedding Similarity (EMB; Muller et al. (2023)) | 70.12 | 51.81 | 16.67 | 74.65 |
| | | **Ours (X-SNS)** | **72.55** | **65.52** | **33.33** | **83.73** |
| **NLI** (XNLI) | 15 | Lang2Vec (L2V; Littell et al. (2017)) | 10.24 | 12.80 | **13.33** | 59.77 |
| | | Lexical Divergence (LEX; Muller et al. (2023)) | **37.73** | 9.73 | 6.67 | 60.19 |
| | | Subword Evenness (SuE; Pelloni et al. (2022)) | 0.04 | 2.75 | 8.89 | 58.45 |
| | | Embedding Similarity (EMB; Muller et al. (2023)) | 23.09 | 27.63 | 8.89 | 63.15 |
| | | **Ours (X-SNS)** | 24.47 | **31.52** | 11.11 | **68.73** |
| **PI** (PAWS-X) | 7 | Lang2Vec (L2V; Littell et al. (2017)) | 68.55 | 62.12 | 47.62 | 86.81 |
| | | Lexical Divergence (LEX; Muller et al. (2023)) | 34.41 | 28.16 | 38.10 | 77.11 |
| | | Subword Evenness (SuE; Pelloni et al. (2022)) | 20.41 | 28.26 | 14.29 | 60.37 |
| | | Embedding Similarity (EMB; Muller et al. (2023)) | 47.12 | 48.92 | 23.81 | 83.51 |
| | | **Ours (X-SNS)** | **73.59** | **64.47** | **52.38** | **89.82** |
| **QA** (TyDiQA) | 8 | Lang2Vec (L2V; Littell et al. (2017)) | 66.27 | 54.76 | **62.50** | 84.52 |
| | | Lexical Divergence (LEX; Muller et al. (2023)) | 18.64 | 29.46 | 20.83 | 72.14 |
| | | Subword Evenness (SuE; Pelloni et al. (2022)) | 39.77 | 36.46 | 0.00 | 73.21 |
| | | Embedding Similarity (EMB; Muller et al. (2023)) | 64.80 | 65.18 | 20.83 | 86.00 |
| | | **Ours (X-SNS)** | **72.58** | **68.15** | 50.00 | **87.95** |

Table 2: Comparison with baselines that rank candidate languages based on their (estimated) suitability as the source. The # column indicates the size of the language pools, which consist of elements used as both the source and target. All the reported scores represent the average performance of each method across all the target languages considered. X-SNS consistently outperforms the baselines across various metrics and tasks, confirming its effectiveness.

languages presented by the evaluated method for selecting the proper source language. Since all the metrics are computed regarding one target language and a set of its candidate languages for XLT, we calculate the scores for every target language and average them across all tested languages, showing the overall effectiveness of each method. Scores are displayed in percentage for better visualization.

## 5 Main Results

### 5.1 Performance by Sub-Network Types

We initially explore the efficacy of our method by evaluating its performance regarding the class of sub-networks it utilizes. Results from all available options are listed in Table 1. Note that for $\mathcal{W}$, we adopt the WikiANN (Pan et al., 2017) version of Wikipedia, making $\mathcal{T}=\mathcal{W}$ for $D$ on NER. [5]

The construction of sub-networks relying on task-specific labeled data ($D=\mathcal{T}$ and $p(y|\mathbf{x}; \boldsymbol{\theta})=\mathcal{T}$) yields somewhat unsatisfactory performance. We

suspect this is partially due to the fact that we decide to use random layers rather than trained ones for classification heads. Furthermore, considering that this direction necessitates annotated data for its implementation, we can infer that it is not advisable to employ it for predicting the optimal source language. On the other hand, we achieve the best result when creating sub-networks through the application of (masked) language modeling based on raw text extracted from task-specific datasets, i.e., $D=\mathcal{T}$ and $p(y|\mathbf{x}; \boldsymbol{\theta})=\mathcal{L}$. This is encouraging in that it is more feasible to collect in-domain raw text than labeled data.[6] Given its cheap requirements and exceptional performance, we regard this option as our default choice in the following sections. Lastly, we conduct tests on deriving sub-networks by exploiting a general corpus, in an effort to expand the potential application scope of the proposed approach ($D=\mathcal{W}$ and $p(y|\mathbf{x}; \boldsymbol{\theta})=\mathcal{L}$). Table 1 reports that in terms of $D$, applying $\mathcal{W}$ mostly performs

---

[5]We employ WikiANN, a processed version of Wikipedia, to reduce costs for preprocessing. We plan to apply our method to other corpora sets in future work.

[6]In XLT setup, acquiring plain text in the target domain is readily feasible. Text in the source language can be obtained from the training set, while text in the target language can be gathered directly from the evaluated data instances themselves.

| Task | NER | | | POS | | | NLI | | | PI | | | QA | | |
|---|---|---|---|---|---|---|---|---|---|---|---|---|---|---|---|
| Approach / Metric | RMSE | Top 1 | NDCG@3 | RMSE | Top 1 | NDCG@3 | RMSE | Top 1 | NDCG@3 | RMSE | Top 1 | NDCG@3 | RMSE | Top 1 | NDCG@3 |
| X-POS (multi) + OLS | 11.97 | 27.45 | 74.09 | **9.40** | **33.33** | **88.32** | 4.18 | 8.89 | 58.08 | **3.80** | 33.33 | 75.15 | 9.04 | 29.17 | 73.50 |
| X-SNS (sing.) + OLS | **8.88** | **39.22** | **78.12** | 9.58 | **33.33** | 83.73 | **3.95** | **11.11** | **68.73** | 4.98 | **52.38** | **89.82** | **8.93** | **50.00** | **87.95** |
| X-POS (multi) + MER | 7.18 | 27.45 | 77.10 | **4.71** | **33.33** | **88.21** | 1.69 | 6.67 | 61.00 | **0.85** | **53.97** | **90.98** | 7.40 | 37.50 | 77.77 |
| X-SNS (sing.) + MER | **5.12** | **39.22** | **78.12** | 5.68 | **33.33** | 83.73 | **1.67** | **11.11** | **68.73** | 1.01 | 52.38 | 89.82 | **5.80** | **50.00** | **87.95** |

Table 3: Evaluation in the framework of XLT regression. There are two regression techniques—ordinary least square (OLS) and mixed effect regression (MER)—and three metrics—RMSE ($\downarrow$), Top 1 ($\uparrow$), and NDCG@3 ($\uparrow$). The results show that the single feature computed by X-SNS can substitute multiple linguistic features, verifying the integrity and impact of our method.

worse than $\mathcal{T}$, highlighting the importance of in-domain data. We leave investigation on the better use of general domain data as future work.

## 5.2 Comparison with Ranking Methods

We here compare our method against baselines that rank candidate languages based on their (estimated) suitability as the source. The results are in Table 2.

Overall, the proposed method demonstrates its superiority across diverse metrics and tasks, thereby confirming its effectiveness for reliably predicting the success of XLT. For instance, it achieves an improvement of 4.6% on average in terms of NDCG@3. On the other hand, L2V generally exhibits inferior performance, although it attains the highest Top 1 accuracy on NLI and QA. We suspect the high scores achieved in those cases are partially attributed to the limited size of the language pools used for the tasks, making the tasks vulnerable to randomness. SuE also shows unsatisfactory performance. This is mainly because SuE lacks a mechanism to reflect task-specific cues, resulting in over-generalized suggestions which are presumably suboptimal. EMB closely resembles our method in terms of both its mechanism and performance. However, X-SNS provides more accurate predictions, and this gap becomes substantial when we assess the practical utility of the predictions, as confirmed in §6.4. In addition, our method opens up avenues for analyzing the structural activation patterns of language models, implying its potential utility for probing. Finally, note that all methods including ours perform poorly in the NLI task. We analyze this phenomenon in Appendix C.

## 5.3 Comparison with Regression Methods

Previously, we focused on evaluating ranking methods. Meanwhile, there is another line of research that seeks to estimate precise XLT scores, going beyond the scope of identifying suitable source languages (de Vries et al., 2022; Muller et al., 2023).

While X-SNS is originally designed as a ranking method, its outcomes can also be utilized as a feature in the regression of XLT scores. We thus delve into the potential of X-SNS in the realm of XLT regression. As baselines, we consider using a set of the multiple features employed in X-POS (de Vries et al., 2022) that include information on language family, script, writing system type, subject-object-verb word order, etc. Following the previous work, we present the results of applying two regression techniques— ordinary least squares (OLS) and mixed-effects regression (MER)— on the features. In our case, we conduct regression relying solely on sub-network similarity. As evaluation indices, the root mean squared error (RMSE) is introduced in addition to ranking metrics (Top 1 and NDCG@3). The points representing the XLT results of all possible pairs of source and target languages are regarded as inputs for the regression process. Table 3 show that X-SNS + OLS surpasses X-POS + OLS in 4 tasks, with the exception of POS tagging for which the X-POS features are specifically devised in prior research. This implies that the single feature provided by X-SNS is sufficient to replace the multiple features proposed in the previous work, emphasizing the richness of information encapsulated in sub-network similarity. X-SNS + OLS also exhibits comparable performance with X-POS + MER, despite its algorithmic simplicity. Furthermore, the effectiveness of our method in score approximation, as evidenced by the RMSE, can be enhanced by incorporating MER (X-SNS + MER).

## 6 Analysis

We present extensive analyses helpful for a deeper understanding of X-SNS. We explore some factors that can have an impact on its performance, such as the sub-network ratio $p$ (§6.1), the base language model (§6.2), and the size of the corpus $D$ (Ap-

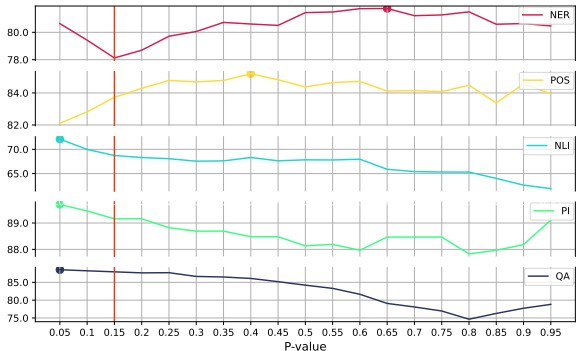

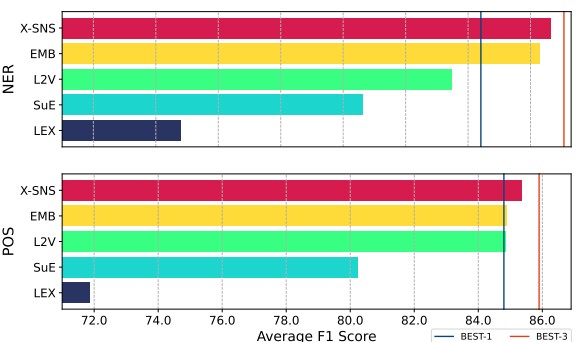

Figure 3: Fluctuations in performance (in NDCG@3) when varying the value of $p$. For token-level tasks (NER, POS), a moderate level of $p$ is optimal, whereas the lowest value for $p$ is the best for other semantic tasks.

Figure 4: Performance on NER and POS tasks in DMT settings, according to the selection of 3 source languages following suggestions from each prediction (or ranking) method. The evaluation metric is the F1 score averaged over all target languages. X-SNS outperforms baselines and achieves exclusive improvement in the POS task.

|  | Pearson | | Spearman | | Top 1 | | NDCG@3 | |
|---|---|---|---|---|---|---|---|---|
|  | EMB | X-SNS | EMB | X-SNS | EMB | X-SNS | EMB | X-SNS |
| **mBERT** | 81.21 | **86.83** | 40.18 | **58.67** | 23.53 | **41.18** | 73.81 | **80.15** |
| **XLM-R** | 78.18 | **78.80** | 49.52 | **58.20** | 31.37 | **39.22** | 76.06 | **78.12** |
| **mT5** | 85.77 | 68.06 | **56.14** | 43.65 | 25.49 | **29.41** | 76.41 | **76.72** |

Table 4: Comparison of the performance of X-SNS and EMB when they are applied on various types of language models.

pendix D). Moreover, we assess the efficacy of the approach in low-resource scenarios (§6.3) and in XLT with multiple source languages (§6.4).

## 6.1 Grid Search on Hyperparameters

In this study, we intentionally fix the value of $p$ as 0.15, following the mask perturbation ratio in language modeling, to minimize an effort in hyperparameter search. We examine how much robust this decision is, checking fluctuations in performance while varying the value of $p$. Figure 3 reveals that the prediction of X-SNS is optimized when utilizing low values of $p$ for high-level tasks (PI, NLI, QA), while moderate values of $p$ are more effective for low-level tasks. However, the gap between the best and worst performance of the method is not substantial, except for the case of QA.

## 6.2 Impact of Base Language Models

While main experiments were concentrated on testing methods with XLM-RoBERTa, our approach can be applied on other multilingual models such as mBERT_Base(Devlin et al., 2019) and mT5_Base(Xue et al., 2021). To validate the robustness of our approach, we replicate our experiments conducted on XLM-R, replacing the language model with mBERT and mT5. We evaluate these variations

on the NER task, comparing the outcomes with those of EMB, a model-based method similar to X-SNS. The experimental results are illustrated in Table 4, which show that X-SNS outperforms EMB when applied on mBERT. Meanwhile, X-SNS and EMB demonstrate comparable performance when evaluated on mT5. While experiments with mT5 illuminate the potential of extending our approach to various types of pre-trained models, we reserve deeper exploration in this direction for future work

## 6.3 Evaluation on Low-Resource Languages

The advantage of XLT is particularly magnified in scenarios where there are constraints on data collection for the target task and language. So far, we have utilized the same language pool for both the source and target languages to facilitate evaluation. However, targets for XLT often in practice encompass languages that were not even participated in the pre-training phase of base models. To validate our method in such scenarios, we perform an experiment on NER with low-resource languages, with the number of data instances for evaluation ranging from 1000 to 5000. The selection consists of 15 languages, three of which are not present in XLM-R pre-training. We observe that in 11 out of 15 cases, X-SNS outperforms using English, achieving an average F1 score improvement of 1.8 points. More details on this experiment are in Appendix E.

## 6.4 XLT with Multiple Source Languages

A prior study (Singh et al., 2019) reported that in XLT, incorporating multiple source languages during training can enhance performance, referring to this setup as "disjoint multilingual train-

ing (DMT)" settings. In Figure 4, our findings also reveal that BEST-3 (XLT based on a mixture of the top 3 source languages) surpasses BEST-1 (trained on the single best source language), indicating that the number of source languages used for training in XLT can fairly impact its performance. However, the vast number of exponentially possible combinations in selecting source languages poses a challenge in searching the optimal one for DMT settings. In this condition, source language prediction (or ranking) methods including X-SNS can come as a rescue to the problem, providing plausible options to be selected. We therefore conduct an experiment on testing the ability of ranking methods in DMT settings. Concretely, we employ the combination of the top 3 source languages as suggested by each method to refine a base model through fine-tuning. In Figure 4, we confirm that X-SNS yields the best performance. Note that the amalgamation of the source languages proposed by X-SNS exhibits exclusive enhancements in the POS task when compared to BEST-1 (the blue vertical line), implying that it is the only option that proves to be truly beneficial in that configuration.

## 7   Conclusion

We present X-SNS, a method that efficiently predicts suitable source languages for XLT with minimal requirements. We identify that XLT performance between two languages has a strong correlation with the similarity of sub-networks for those languages, and propose a way of exploiting this fact in forecasting the success of XLT. X-SNS demonstrates exceptional versatility and robustness, verifying impressive performance across various environments and tasks. We believe that our suggestions and findings have the potential to contribute to the exploration of the inner workings of multilingual language models, leaving it as future work.

## Limitations

**Need for exploration on sub-network construction strategies**   While our proposal focuses on leveraging Fisher Information as a source of information for constructing sub-networks, it is important to note that there exist other alternatives that can be utilized to achieve the same objective. We argue our main contribution lies in the fact that we introduce the framework of using sub-networks for predicting cross-lingual transfer (XLT) performance and that the utilization of Fisher Information

serves as a notable example of implementing this framework. Consequently, we anticipate that researchers within the community will collaborate to enhance the framework in the near future by developing more effective methodologies for extracting sub-networks from a base model.

**Limited availability of multilingual evaluation datasets**   Despite our diligent efforts to evaluate our approach across multiple configurations, including five evaluation tasks, it is crucial to acknowledge the scarcity of multilingual datasets that can effectively test the practical efficacy of cross-lingual transfer across the vast array of over a hundred languages. As a result of this limitation, we were constrained to focus our analysis on only a few selected tasks, such as NER and POS tagging. We advocate for the development of additional multilingual datasets that encompass a broad spectrum of languages. Such comprehensive datasets would greatly facilitate the evaluation of methods for XLT, including our own approach.

**Relatively heavy computational cost**   Regarding computational resources, our method necessitates relatively higher storage capacity to accommodate gradient information alongside model parameters. Nonetheless, this issue can be largely alleviated by reducing the amount of data required for the algorithm's execution., e.g., $|D| = 256$ or $1024$. Furthermore, our algorithm eliminates the need for directly tuning a variety of models for testing the success of XLT, making it eco-friendly.

## Ethics Statement

In this work, we perform experiments using a suite of widely recognized datasets from the existing literature. As far as our knowledge extends, it is highly unlikely for ethical concerns to arise in relation to these datasets. However, since the proposed method has the potential to be applied to address new datasets in low-resource languages, it is crucial to approach data collection with careful consideration from an ethical standpoint in those cases.

## Acknowledgements

This work was supported by Hyundai Motor Company and Kia. This work was supported by Institute of Information & communications Technology Planning & Evaluation (IITP) grant funded by the Korea government(MSIT) (No.2020-0-01373, Artificial Intelligence Graduate School

Program (Hanyang University)). This work was supported by the National Research Foundation of Korea(NRF) grant funded by the Korea government(MSIT) (No.2022R1F1A1074674).

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

# Appendix

## A  Details on Tasks and Datasets

**Named Entity Recognition (NER)** is a task that involves the classification of pre-defined entity types for each name mention within a sentence. We employ WikiANN (Pan et al., 2017) for evaluation on named entity recognition. This dataset covers 282 languages existing in Wikipedia, but we curtail the number of languages to be considered as a source to 17 based on two criteria: (1) the selected language should contain over 20K data instances, and (2) it was evaluated in the previous study (Lauscher et al., 2020). The final list of the candidate languages is as follows: Arabic (AR), Chinese (ZH), English (EN), Finnish (FI), French (FR), Greek (EL), Hebrew (HE), Indonesian (ID), Italian (IT), Japanese (JA), Korean (KO), Russian (RU), Spanish (ES), Swedish (SV), Thai (TH), Turkish (TR), and Vietnamese (VI). The micro F1 score is used as a metric.

**Part-of-Speech Tagging (POS)** is a sequence-labeling task that assigns the most appropriate part-of-speech tag to each word in a sentence. For this task, we adopt a variation of Universal Dependencies 2.8 (Zeman et al., 2021) processed by de Vries et al. (2022). We consider 20 languages as candidates for transfer, whose number of training data is over 4K: Bulgarian (BG), Chinese (ZH), Dutch (NL), English (EN), Finnish (FI), French (FR), Hindi (HI), Indonesian (ID), Italian (IT), Japanese (JA), Korean (KO), Norwegian (NO), Polish (PL), Portuguese (PT), Russian (RU), Slovakia (SK), Spanish (ES), Swedish (SV), Turkish (TR), and Ukrainian (UK). We use micro F1 for evaluation.

**Natural Language Inference (NLI)** aims to classify the entailment relationship between two pieces of text. We choose the XNLI (Conneau et al., 2018) dataset for use, covering 15 languages: Arabic (AR), Bulgarian (BG), Chinese (ZH), English (EN), French (FR), Greek (EL), German (DE), Hindi (HI), Russian (RU), Spanish (ES), Swahili (SW), Thai (TH), Turkish (TR), Urdu (UR), and Vietnamese (VI). It comprises 390K instances of machine-translated training data for each language, along with 5K instances of human-translated test data per language. For evaluation, accuracy is used.

**Paraphrase Identification (PI)** intends to deter-

mine whether a given pair of sentences is a paraphrase of one another. We use the PAWS-X dataset (Yang et al., 2019), spanning 7 languages: Chinese (ZH), English (EN), French (FR), German (DE), Japanese (JA), Korean (KO), and Spanish (ES). it contains 50K machine-translated instances in the training set, as well as 2K human-translated ones in the test set. Accuracy is used for evaluation.

**Question Answering (QA)**, more specifically, reading comprehension aims to retrieve a continuous span of characters in the context that answers the given question. We utilize TyDiQA (Clark et al., 2020) which provides a non-translated dataset that consists of typologically diverse 9 languages: Arabic (AR), Bengali (BG), English (EN), Finnish (FI), Indonesian (ID), Korean (KO), Russian (RU), Swahili (SW), Telugu (TE). To train the base model, we split the original training set into 80% and 20% for training and validation respectively. For stable evaluation, we exclude Korean which has the smallest number of training data. The F1 score is used as a metric.

## B  Detailed Specification on Baselines

**Lang2Vec (L2V)**  We incorporate all features belonging to the "typological" category in the L2V utility: syntactic, phonological, and inventory vectors with the kNN predicted version to fill in missing elements in the vectors. We concatenate all these vectors to express the typological characteristics of a language as a single vector. We then calculate a cosine similarity between two representative vectors of each source and target language, and use it as a predictor for the efficacy of XLT.

**Lexical Divergence (LEX)**  Following Muller et al. (2023), we define the lexical divergence between source and target languages as the Jensen-Shannon Divergence (JSD) of the sub-word uni-gram distributions of each language:

$$\text{LEX}_{(s,t)} = \text{JSD}(D_s, D_t),$$

where $D_l$ corresponds to the sub-word uni-grams distribution of the language $l$.

**Subword Evenness (SuE)**  As another instance of statistical approaches, we consider the SuE score proposed by Pelloni et al. (2022). We first construct a source language corpus for each task utilizing its training data. And then we derive the SuE score

per language ($s$) and task as follows:

$$\text{SuE}_s = 180° - |\arctan 1| - |\arctan k|,$$

where $|\arctan 1|$ and $|\arctan k|$ are the lower angles of the triangle that describes the distribution of points that correspond to each word in a corpus, whose x-axis is about the length of the word and y-axis represents unevenness scores. For a more detailed explanation on this algorithm, we recommend readers to refer to Pelloni et al. (2022).

**Embedding Similarity (EMB)** Following Muller et al. (2023), we calculate the cosine similarity between embedding vectors of the source and target language as follows:

$$\text{EMB}_{(s,t)} = \frac{\mathbf{e}_s \cdot \mathbf{e}_t}{\|\mathbf{e}_s\|\|\mathbf{e}_t\|}.$$

Note that $\mathbf{e}_s$, $\mathbf{e}_t$ are embedding vectors representing the source and target languages respectively. To derive $\mathbf{e}$, we compute a mean-pooled vector by averaging token embeddings from the last layer of a language model. We then compute $\mathbf{e}$ by taking an average of all those mean vectors derived from 1024 data examples. We choose 1024 as the number of examples for EMB to guarantee a fair comparison with our method.

## C Performance Gap in XLT across Tasks

This section explains the reasons why experimental results on NLI were noisy for all considered methods. First, it turns out that predicting appropriate source languages for both NLI and PI tasks is tricky because the variations of XLT scores across different source languages on those tasks are minimal, as shown in Figure 5. Secondly, in the context of NLI, we encounter instances where the performance of XLT surpasses that of conventional fine-tuning on the target language itself, raising questions about the suitability of the XNLI dataset as a reliable benchmark for evaluating the effectiveness of XLT.

## D Impact of $|D|$ on Performance

In this part, we examine the fluctuations in performance observed within our method as we modify the value of $|D|$. We sample the size of data examples in the set {64, 128, 256, 512, 1024, 10000}. As depicted in Figure 6, the NDCG@3 scores converges when utilizing more than 1024 examples, justifying our decision of using this number in our main experiments. Note also that the data size of nearly 256 can be a reasonable choice for efficiency without compromising performance.

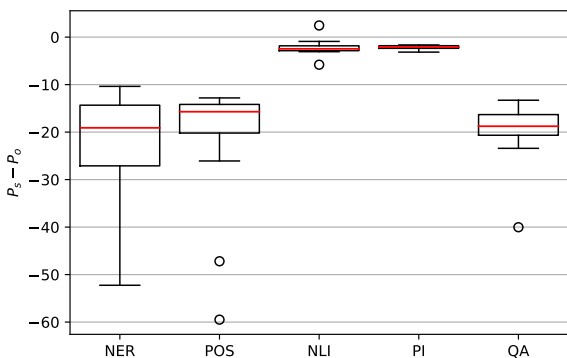

Figure 5: Visualization of the distributions of performance differences ($P_s - P_o$) caused by the selection of source languages. $P_s$ corresponds to XLT performance when fine-tuned on the source language $s$ while $P_o$ is the task performance shown when the language model is task-specifically trained on the target language itself. Therefore, the distributions of $P_s - P_o$ demonstrate the expected amounts of performance drop when we substitute the target language with each candidate source language. Note that there exist even positive gaps ($P_s - P_o > 0$) in the NLI task, implying that the XNLI dataset that represents the NLI task might be not suitable for reliable evaluation of performance in XLT settings.

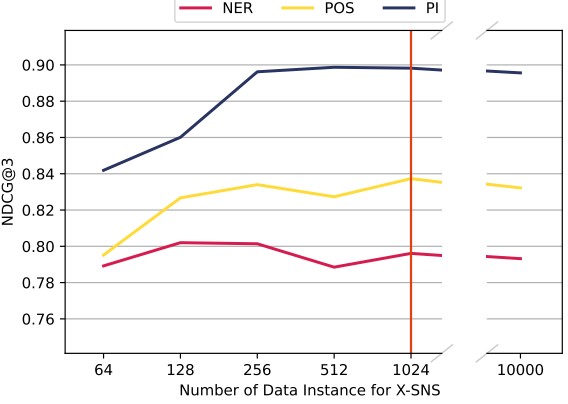

Figure 6: The NDCG@3 scores plotted based on the number of data instances ($|D|$) used to construct a subnetwork in the condition of $D = \mathcal{T}$ and $p(y|\mathbf{x}; \boldsymbol{\theta}) = \mathcal{L}$. The vertical orange line indicates the number of data (1024) applied for our main experiments in this paper.

## E Evaluation on Low-Resource Languages

In our experiment, we utilize a set of 17 resource-rich languages from the WikiANN dataset as source languages. For target languages, we consider 15 low-resource languages: Afrikaans (AF), Breton (BR), Kurdish (CKB), Western Frisian (FY), Irish (GA), Hindi (HI), Icelandic (IS), Kazakh

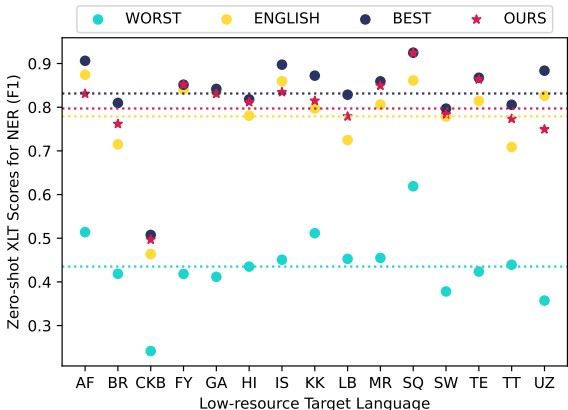

Figure 7: This figure illustrates that zero-shot XLT performance for low-resource target languages on NER can vary depending on the choice of source languages. Our method outperforms using English in 11 out of 15 cases. The horizontal lines are the average score for each case.

(KK), Luxembourgish (LB), Mauritania (MR), Albanian (SQ), Swahili (SW), Telugu (TE), Tatar (TT), Uzbek (UZ). These languages are selected from the same dataset whose number of data (for evaluation) ranges from 1000 to 5000. We choose the pseudo-optimal source language by identifying the one that exhibits the highest sub-network similarity for each target language. From the results in Figure 7, we find out that we can obtain better results when conducting XLT with the source languages recommended by our method, compared to the case of using English as the source language, resulting in an average improvement of 1.8 points.