# OpenReview forum: "X-SNS: Cross-Lingual Transfer Prediction through Sub-Network Similarity"
_EMNLP/2023/Conference — EMNLP 2023 Findings_

### Official Review · Reviewer_uSgC · 2023-08-03

**Soundness:** 3

**Excitement:**

3: Ambivalent: It has merits (e.g., it reports state-of-the-art results, the idea is nice), but there are key weaknesses (e.g., it describes incremental work), and it can significantly benefit from another round of revision. However, I won't object to accepting it if my co-reviewers champion it.

**Paper Topic And Main Contributions:**

This paper focuses on the zero-shot cross-lingual transfer method, which fine-tunes multilingual language models on the source language, and then evaluates on the target language.

Instead of using English as the source language or using features which are costly to obtain for choosing source language, this paper proposes sub-network similarity to choose source language. Specifically, if a source language gets a higher sub-network similarity to the target language, it would be regarded as a better source language while conducting zero-shot cross-lingual transfer for the target language.

The experimental results show that sub-network similarity can choose better source languages compared to previous methods on a diverse set of downstream tasks.

**Questions For The Authors:**


A.Line 18, what do ‘external resources’ refer to?

B.In Figure 2, Is Step 3 (top p%) necessary? Is it possible to obtain the similarity based on the outputs of Step 2?

C.Line 275-279, does previous paper for choosing source languages require ‘the need for extra training of base models’, Please give the citation of previous work. Does ‘doing so’ refer to ‘considers the relative significance’?

D.Line 287, why is the general domain set as Wikipedia instead of CC-100? CC-100 is the corpus used for training XLM-R.

E.Line 304, what is random classifier?

F.Line 340-345, lang2vec provides six metrics for measuring linguistics similarity across languages. Do this paper use the ‘feature’ one? If yes, please also compare with other 5 separate metrics. Lin et al. 2017 show that ‘feature’ is not best one for predicting performance of cross-lingual transfer.

G.Table 2, 3, why are different baselines used?

H.Line 544-548, can X-SNS deal with the case that combining a new language might worse the results?

I.Line 550, while considering 3 source languages, are there the same amount of data used for fine-tuning as considering single source language?

J.Are the conclusions the same if X-SNS is applied in some other multilingual language models, such as large language models, e.g., Bloom, and speech-based models, e.g., XLS-R?

K.Analysis on fail cases can show the shortages of X-SNS and provide insights for reaching the upperbound (best one).

L.Is the fine-tuning on the source language conducted in the whole multilingual model or the shared sub-network? Analysis on how the sub-network change during fine-tuning can better explain the success/fail cases of X-SNS.

M.Please provide the time cost of X-SNS.

N.How is the source language set defined?


**Reasons To Accept:**

- This paper presents a novel language similarity measure for zero-shot improving cross-lingual transfer.
- The experimental results show that the proposed method outperforms previous methods.

**Reasons To Reject:**

- Although the paper is well organized, some details are not clear and some experiments are missing (see questions in details)

**Reproducibility:**

4: Could mostly reproduce the results, but there may be some variation because of sample variance or minor variations in their interpretation of the protocol or method.

**Reviewer Confidence:**

5: Positive that my evaluation is correct. I read the paper very carefully and I am very familiar with related work.

---

> ### Author Rebuttal · Authors · 2023-08-29
>
> We are grateful for the dedication and effort you've invested in grasping the nuances of our work.
>
> In the following, let us address your questions.
>
> Through this process, we anticipate further improving the quality of our work.
>
> ---
>
> **A. Line 18, what do ‘external resources’ refer to?**
>
> ---
>
> The term ‘external resources’ refers to linguistic databases employed by previous work, such as the ASJP database in de Vries et al. (2022)[1] and the WALS database in Muller et al. (2023)[2].
>
> In our revision, we will elucidate this point, emphasizing that our work depends on plain text instead of pre-defined features, which may not always be available for every language.
>
> ---
>
> **B. In Figure 2, Is Step 3 (top p%) necessary? Is it possible to obtain the similarity based on the outputs of Step 2?**
>
> ---
>
> Your suggestion is possible in principle and also reasonable.
>
> Therefore, in the next revision, let us consider conducting extra experiments on the proposed variant (removing Step 3).
>
> However, we would like to highlight that Step 3 is vital for keeping our language vectors as simple as possible (i.e., binary masks or subtrees).
>
> This is desired because the language vectors would be frequently used for computing the similarity between every possible language pair, causing computational burdens.
>
> If we manage to use binary features, we can readily rely on bitwise AND and OR operations, which are more efficient than doing the dot products of real-valued vectors.
>
> That’s the main reason why we decided to introduce Step 3.
>
> On the other hand, in terms of controlling the $p$-value, we presented an analysis on this topic in Section 6.1.
>
> Please take a look at the section.
>
> ---
>
> **C. Line 275-279, does previous paper for choosing source languages require ‘the need for extra training of base models’, Please give the citation of previous work. Does ‘doing so’ refer to ‘considers the relative significance’?**
>
> ---
>
> The phrase "doing so" refers to the entire process for calculating sub-networks.
>
> Let us revise the writing in that part if our intention was not clearly delivered.
>
> In the paragraph, we aimed to emphasize that in the process of constructing sub-networks,  we only compute gradients and do not actually update the weights of the model, unlike previous research (Xu et al. 2021)[3].
>
> We will append appropriate citations to the proper location.
>
> ---
>
> **D. Line 287, why is the general domain set as Wikipedia instead of CC-100? CC-100 is the corpus used for training XLM-R.**
>
> ---
>
> Wikipedia is frequently used as a base for pre-training numerous language models.
>
> Furthermore, it stands as a pivotal resource in building datasets for a wide range of tasks.
>
> This underscores our rationale for choosing Wikipedia to represent the general domain.
>
> Indeed, we recognize the merit of your suggestion.
>
> In our subsequent revision, we will contemplate evaluating our approach using CC-100 in lieu of Wikipedia.
>
> Thanks again for your comment.
>
> ---
>
> **E. Line 304, what is random classifier?**
>
> ---
>
> As we mentioned in Section 3, we need a classifier to calculate a task-specific gradient in the case of $p(y|x;\theta) = T$.
>
> For this purpose, we add a randomly initialized classifier on top of the pre-trained model and compute the gradient.
>
> However, when $p(y|x;\theta)=W$, the task that contributes to computing the gradient is an MLM task, so a pre-trained LM head is used instead of the random classifier.
>
> ---
>
> **F. Line 340-345, lang2vec provides six metrics for measuring linguistics similarity across languages. Do this paper use the ‘feature’ one? If yes, please also compare with other 5 separate metrics. Lin et al. 2017 show that ‘feature’ is not best one for predicting performance of cross-lingual transfer.**
>
> ---
>
> To the best of our knowledge, lang2vec offers six ‘metrics’ (including what you referred to as the 'feature'), and all of these are distance-based.
>
> In our preliminary experiments, we discovered that instead of directly leveraging the distance-based ‘metrics’, it is more competitive for the baseline to manually compute cosine similarities based on the ‘feature vectors’ also provided by lang2vec.
>
> Therefore, we decided to adopt the latter approach (manual computation of similarity), as it guarantees the improved performance of the baseline (lang2vec).
>
> ---
>
> **G. Table 2, 3, why are different baselines used?**
>
> ---
>
> For the comparative analysis of our research, we categorized baselines into two groups following the literature.
>
> Specifically, the first group concentrates on recommending languages anticipated to transfer effectively, while the second group directly predicts the precise transfer scores calculated when the transfer is actually executed.
>
> Table 2 showcases the performance of approaches from the first group. The primary objective of these approaches is to provide features that predict the compatibility between the source and target languages.
>
> Conversely, Table 3 lists the performance of methods from the second group. These methods utilize a regression approach to predict the actual transfer score for the source language.
>
> In summary, our aim was to compare baselines within their respective groups, which is why we presented the results for each group in distinct tables.
>
> ---
>
> **H. Line 544-548, can X-SNS deal with the case that combining a new language might worsen the results?**
>
> ---
>
> As you highlighted, increasing the number of source languages doesn't necessarily yield positive outcomes.
>
> Still, we assert that our method has the capability to discerningly select languages that reduce potential negative effects, highlighting its advantage over other baselines.
>
> To substantiate this claim with tangible results, we outline below the frequency of instances where the 3-language-tuning setting, as chosen by each method, leads to a decline in performance relative to that of Best-1.
>
> |         | X-SNS | EMB | L2V | SuE | LEX |
> |---------|:-------:|:-----:|:-----:|:-----:|:-----:|
> | NER(17) | 2     | 3   | 4   | 11  | 16  |
> | POS(20) | 5     | 9   | 9   | 18  | 18  |
>
> `Table` The number in parentheses next to the task means the number of target languages, and the elements in the table correspond to the number of cases causing negative effects.
>
> We will incorporate the findings from this additional experiment to further bolster our claim that our method remains effective, even under extreme conditions.
>
> Thank you for suggesting we consider this dimension.
>
> ---
>
> **I. Line 550, while considering 3 source languages, are there the same amount of data used for fine-tuning as considering single source language?**
>
> ---
>
> Yes, to clarify:
>
> When using multiple source languages (for instance, 3), we allocate an equal amount of data ($|D|$) for each language.
>
> This results in a total data size of $3 \times |D|$ for the combined training.
>
> Let's assume the number of epochs used for the multi-language training is $e$.
>
> Now, when calculating the Best-1 score using a single language, we train with data of size $|D|$ but over $3 \times e$ epochs.
>
> This ensures that the total number of training steps remains consistent across both scenarios.
>
> ---
>
> **J. Are the conclusions the same if X-SNS is applied in some other multilingual language models, such as large language models, e.g., Bloom, and speech-based models, e.g., XLS-R?**
>
> ---
>
> We believe that your suggestion reveals a future direction we should pursue, i.e., attempting to extend the application of our approach to the case of pre-trained decoders.
>
> Nevertheless, we assert that our experimental setup is comprehensive, particularly given that cross-lingual transfer has predominantly been explored within the context of encoder-based models.
>
> Thank you again for your constructive suggestion for future avenues.
>
> ---
>
> **K. Analysis on fail cases can show the shortages of X-SNS and provide insights for reaching the upperbound (best one).**
>
> ---
>
> Thanks for the suggestion.
>
> We also believe that it is important to show both the strengths and weaknesses of our method.
>
> We tested with various target languages, and most of the languages produced successful predictions, but there were some failed cases.
>
> To be specific, we found that in three low-resource languages, i.e., Afrikaans, Uzbek, and íslenska, the transfer performance of our method was worse than using English.
>
> For more details, please refer to Figure 1 and Figure 7 in addition to Appendix F.
>
> ---
>
> **L. Is the fine-tuning on the source language conducted in the whole multilingual model or the shared sub-network? Analysis on how the sub-network change during fine-tuning can better explain the success/fail cases of X-SNS.**
>
> ---
>
> For your better interpretation, we should first make it clear that our method does not involve fine-tuning.
>
> We only calculate the gradient for each language as a source of information.
>
> However, we do not undertake any optimization steps (like gradient descent) based on this gradient.
>
> The rationale behind this approach is that our objective is to identify a source language optimal for transfer without actively tuning the model.
>
> If we were to update the model, it would introduce a contradiction to our intended method.
>
> Indeed, your recommendation to examine the dynamics of sub-network alterations is intriguing.
>
> We believe it can be integrated into the source tuning phase, either to analyze patterns by language or to enhance the elucidation of our methodology.
>
> ---
>
> **M. Please provide the time cost of X-SNS.**
>
> ---
>
> Extracting a feature for one language takes 37.14 seconds, while comparing features requires 3.81 seconds.
>
> When computing the language feature with 1024-shot data, the gradient calculation consumes approximately 20.74 seconds.
>
> However, this duration can be shortened to 6.32 seconds if we reduce the dataset size for feature calculation to 256.
>
> Notably, this approach has a time-efficiency advantage.
>
> Instead of undergoing the actual source training process, which takes 2 minutes and 31 seconds per epoch, we can identify the source language in a fraction of that time, yielding significant savings in terms of time cost.
>
> ---
>
> **N. How is the source language set defined?**
>
> ---
>
> For alignment with existing studies, we curated our language set to encompass languages that prior research has employed for transfer language prediction.
>
> A comprehensive breakdown of the language set composition can be found in Appendix A.
>
> ---
>
> Once again, we deeply appreciate your diligent efforts and the insightful questions you've brought forward.
>
> We hope that our responses address your questions satisfactorily.
>
>
> >**Reference**
> >
> > [1] [Make the Best of Cross-lingual Transfer: Evidence from POS Tagging with over 100 Languages](https://aclanthology.org/2022.acl-long.529) (de Vries et al., ACL 2022)
> >
> > [2] [Languages You Know Influence Those You Learn: Impact of Language Characteristics on Multi-Lingual Text-to-Text Transfer](https://proceedings.mlr.press/v203/muller23a.html) (Muller et al., PMLR 2023)
> >
> > [3] [Raise a Child in Large Language Model: Towards Effective and Generalizable Fine-tuning](https://aclanthology.org/2021.emnlp-main.749) (Xu et al., EMNLP 2021)

---

### Official Review · Reviewer_TZ8W · 2023-08-05

**Soundness:** 4

**Excitement:**

3: Ambivalent: It has merits (e.g., it reports state-of-the-art results, the idea is nice), but there are key weaknesses (e.g., it describes incremental work), and it can significantly benefit from another round of revision. However, I won't object to accepting it if my co-reviewers champion it.

**Missing References:**

References are adequate


**Paper Topic And Main Contributions:**

In this paper, the authors evaluated the impact of sub-network similarity on cross-lingual transfer prediction.
The authors present their results for five tasks (NER, POS, NLI, PI, and QA) from the XTREME benchmark.
This paper has a clear contribution to cross-lingual prediction and will undoubtedly impact future research.
This paper conducts a range of experiments studying the proposed objectives.



**Reasons To Accept:**

Strengths:

The paper is well-written and easy to follow.

An interesting idea that is relevant to the NLP community.

Conducted experiments on five datasets.

Extensive experiments show the utility of the proposed method.


**Reasons To Reject:**


It must be clarified if the proposed method holds across a wide range of tasks.

**Reproducibility:**

4: Could mostly reproduce the results, but there may be some variation because of sample variance or minor variations in their interpretation of the protocol or method.

**Reviewer Confidence:**

3: Pretty sure, but there's a chance I missed something. Although I have a good feel for this area in general, I did not carefully check the paper's details, e.g., the math, experimental design, or novelty.

---

> ### Author Rebuttal · Authors · 2023-08-28
>
> We appreciate your effort in writing a careful review for our paper, which we believe would be a cornerstone for improving the quality of our submission.
>
> You have pointed out some positive points of our work: (1) the paper is well-written and easy to follow, (2) proposed an interesting idea that is relevant to the NLP community, (3) conducted experiments on five datasets, which are extensive enough to demonstrate the utility of the proposed method.
>
> On top of that, we would like to relieve your remaining concerns about our work.
>
> ---
>
> **1. It must be clarified if the proposed method holds across a wide range of tasks.**
>
> ---
>
> On the journey of doing this research, we tried our best to evaluate our method on as many tasks as possible.
>
> We believe that this is one of the reasons why you mentioned that our experimental settings are extensive.
>
> In addition, we would like to inform that unfortunately, there exist a limited number of datasets suitable for testing the ability of cross-lingual transfer.
>
> Probably due to this reason, previous studies mostly considered only a subset of the tasks we adopted for evaluation. For instance, Pelloni et al. (2022)[1] tested their method only on a single masked language modeling task, de Vries et al. (2022)[2] handled only one POS task, and Muller et al. (2023)[3] conducted experiments with three tasks (NER, NLI, and QA).
>
> Given this background, we can argue with confidence that our work definitely covered a wide range of tasks, validating the extensive effectiveness of our approach.
>
> >**Reference**
> >
> > [1] [Subword Evenness (SuE) as a Predictor of Cross-lingual Transfer to Low-resource Languages](https://aclanthology.org/2022.emnlp-main.503) (Pelloni et al., EMNLP 2022)
> >
> > [2] [Make the Best of Cross-lingual Transfer: Evidence from POS Tagging with over 100 Languages](https://aclanthology.org/2022.acl-long.529) (de Vries et al., ACL 2022)
> >
> > [3] [Languages You Know Influence Those You Learn: Impact of Language Characteristics on Multi-Lingual Text-to-Text Transfer](https://proceedings.mlr.press/v203/muller23a.html) (Muller et al., PMLR 2023)

---

### Official Review · Reviewer_39c7 · 2023-08-11

**Soundness:** 3

**Excitement:**

2: Mediocre: This paper makes marginal contributions (vs non-contemporaneous work), so I would rather not see it in the conference.

**Paper Topic And Main Contributions:**

This work proposes a method to utilize sub-network similarity between two languages as a proxy for predicting the compatibility of the languages in cross-lingual transfer.

**Questions For The Authors:**

1. Values missed in Table 1?


**Reasons To Accept:**

The motivation for using sub-network similarity is reasonable. The evaluations are comprehensive. The results are decent.

**Reasons To Reject:**

1. missing results MER in X-SNS and testing its effectiveness?
2. The p-value in NER and POS change greatly as the p-value change. So maybe present some results of X-SNS with diverse p-values in a table for analysis? An efficient selection method for p-value is needed because searching for p-value mat becomes a major cost of applying X-SNS.
3. Lack of details about applying wiki data as a corpus.
4. X-SNS seems to work poorly in mT5 in Person and Spearman metrics, this requires explanation because if X-SNS works poorly in generation-based models, its application will become much shallower. And I believe more experiments for generation-based backbones may be better.

**Reproducibility:**

3: Could reproduce the results with some difficulty. The settings of parameters are underspecified or subjectively determined; the training/evaluation data are not widely available.

**Reviewer Confidence:**

3: Pretty sure, but there's a chance I missed something. Although I have a good feel for this area in general, I did not carefully check the paper's details, e.g., the math, experimental design, or novelty.

---

> ### Author Rebuttal · Authors · 2023-08-28
>
> We are grateful for taking your valuable time to give constructive and fruitful opinions on our work.
>
> We appreciate the opportunity to address your questions, clarify any misconceptions, and alleviate your concerns.
>
> By doing so, we hope to foster a constructive discussion on the suggestions you've provided to enhance our paper.
>
> ---
>
> **1. Missing MER results for X-SNS in Table 3**
>
> ---
> Thanks for pointing this out.
>
> In the draft, we reported only the performance of the combination of X-SNS and OLS (Ordinary Least Square), not considering X-SNS + MER (Mixed Effect Regression).
>
> Our decision was grounded on two facts: (1) X-SNS + OLS already achieved performance comparable to baselines; and (2) we thought the cost of applying MER (it is much more expensive than that of OLS) outweighs the benefits, hurting the efficiency of our algorithm whose major goal is to predict transfer scores in an efficient manner.
>
> Nonetheless, we admit that it is also desired to evaluate the upper bound of X-SNS’s performance by combining it with MER.
>
> To this end, we have conducted an additional experiment to verify the effectiveness of X-SNS in the suggested setting, whose results (metric: RMSE; lower is better) are listed in the table below.
>
> Our results confirm that X-SNS + MER significantly outperforms X-SNS + OLS.
>
> Furthermore, X-SNS + MER surpasses X-POS + MER in 3 out of 5 tasks.
>
> We will include this outcome in the next version of our draft to emphasize the competence of our approach.
>
> We thank again Reviewer 1 for proposing this additional experiment.
>
> | Method (Metric: RMSE)  | NER   | POS  | NLI  | CLS  | QA   |
> |----------|-------|------|------|------|------|
> | XPOS+OLS | 11.97 | 9.40 | 4.18 | 3.80 | 9.04 |
> | XPOS+MER | 7.18  | 4.71 | 1.69 | 0.85 | 7.40 |
> | XSNS+OLS | 8.88  | 9.58 | 3.95 | 4.98 | 8.93 |
> | XSNS+MER (newly added) | 5.12  | 5.68 | 1.67 | 1.01 | 5.80 |
>
> ---
>
> **2. The performance of X-SNS (for NER and POS) seems changing greatly as the p-value change. Is there any report about X-SNS’s performance with diverse p-values? An efficient selection method for p-value might be needed because searching for p-value can become a major cost of applying X-SNS.**
>
> ---
> As we mentioned in the paper, we intentionally fixed the value of $p$ as 0.15, following the mask perturbation ratio in language modeling, to minimize an effort in hyperparameter search.
>
> We first would like to highlight the fact that although we did not take much care about hyperparameter optimization, X-SNS outperformed competitive baselines.
>
> In fact, as illustrated in Figure 3, our choice of 0.15 for $p$ was even the worst for the NER task.
>
> Nevertheless, X-SNS won over all the baselines considered in our work for the task, which we believe is a promising result.
>
> Moreover, in Section 6.1, we allocated a dedicated section for examining the impact of hyperparameters.
>
> In the subsection, we reported that the performance of X-SNS (metric: NDCG@3) remains largely robust concerning the choice of the $p$-value.
>
> For more explanation, let us introduce another table below.
>
> This table represents the gaps between the upper and lower bounds of NDCG@3 (in the first row), in addition to the gaps between the cross-lingual transfer scores of the source languages (in the second row), while varying the $p$-value.
>
> From the table, we have discovered that our method is generally robust to the selection of $p$, with exceptions in the NLI and QA tasks in terms of NDCG@3 (in the first row).
>
> However, we’d like to argue that these exceptions can be further mitigated when the performance is evaluated with respect to the transfer scores of the chosen source languages, as we can see in the second row of the table.
>
> As the main objective of our approach is to select the most appropriate **single** source language (which is expected to record the highest transfer score) rather than choosing 3 candidates (as measured in NDCG@3), we believe that the setting for the second row is more practical and that the differences shown in the second row are acceptable enough.
>
> We will report this extra investigation in the revision, along with a discussion on the positive and negative impacts of hyperparameter search for our method.
>
> | Max-Min | NER  | POS  | NLI   | CLS  | QA    |
> |---------|------|------|-------|------|-------|
> | NDCG@3  | 3.62 | 3.09 | 10.20 | 1.88 | 13.89 |
> | Transfer Score | 2.27 | 0.47 | 1.44  | 0.72 | 3.81  |
>
> ---
>
> **3. Lack of details about applying wiki data as a corpus.**
>
> ---
>
> On Line 408-410 (in Section 5.1), we already mentioned that we utilized the WikiANN version of Wikipedia.
>
> However, let us specify this fact more clearly on Section 4.3 (Configurations and Metrics) in the next revision.
>
> On the other hand, the reason why we adopted WikiANN as a source of raw text is that it is used for constructing one of our tasks, i.e., NER.
>
> Therefore, by adopting WikiANN (only raw text, without labels for language modeling), we attempted to circumvent the cost needed for preprocessing a large amount of Wikipedia data.
>
> In the revision, we will conduct additional experiments to evaluate the effect of such a corpus.
>
> ---
>
> **4. X-SNS seems to work poorly in mT5 in Person and Spearman metrics, this requires explanation because if X-SNS works poorly in generation-based models, its application will become much shallower. And I believe more experiments for generation-based backbones may be better.**
>
> ---
>
> First of all, we would like to highlight that we mainly focused on evaluating our method with pre-trained encoder models, following the wisdom of the literature that cross-lingual transfer is generally more feasible for the tasks of natural language understanding, rather than generation.
>
> In this sense, relying on encoder models was a reasonable choice, considering that most of the tasks we evaluated can be classified as NLU tasks.
>
> An extension of the paradigm of cross-lingual transfer to generation tasks, with the aid of encoder-decoder and decoder models, is actually out of the scope of this work, and in fact, we are preparing to initiate research on this topic as a following study of this submission.
>
> Nonetheless, in the draft, we attempted to test our approach with encoder-decoder models, such as mT5, as a preliminary step of such an extension.
>
> We admit that it is probable that mT5 is not the best option for the problem setting discussed in this work.
>
> Therefore, we have a plan to conduct extra work that contributes to an extended application of our approach to decoder models (maybe for generation), as a separate future work.
>
> ---
>
> **5. Values missed in Table 1?**
>
> ---
>
> As answered in Q3, we used the raw text part of WikiANN as a Wikipedia dataset.
>
> As it is also a dataset for NER at the same time (when its labels are used together), the datasets for the two different settings (task-specific vs. language modeling) become identical, and that’s why we reported only one set of values to remove repetition.
>
> We mentioned this fact on **Line 410** in the draft.

---

### Meta-Review · Area_Chair_TAJw · 2023-09-19

**Recommendation:** 4

**Metareview:**

This work proposes and studies a method to utilize sub-network similarity between two languages for predicting the source language in cross-lingual transfer. Overall, the reviewers agree that the sub-network similarity is novel for the selection of languages for cross-lingual transfer. Some concerns regarding the clarity and motivation for the proposed framework (which is depicted in Figure 2) and the proposed approach do not work well in some settings such as when using mT5. In summary, the AC believes that the usage of sub-network similarity in choosing the optimal source language is interesting but also recommends addressing the clarity in details and limitations pointed out by reviewers in the next version of the paper.

---

### Decision · Program_Chairs · 2023-10-07

**Decision:**

Accept-Findings

**Comment:**

This work proposes and studies a method to utilize sub-network similarity between two languages for predicting the source language in cross-lingual transfer. Overall, the reviewers agree that the sub-network similarity is novel for the selection of languages for cross-lingual transfer. Some concerns regarding the clarity and motivation for the proposed framework (which is depicted in Figure 2) and the proposed approach do not work well in some settings such as when using mT5. In summary, the AC believes that the usage of sub-network similarity in choosing the optimal source language is interesting but also recommends addressing the clarity in details and limitations pointed out by reviewers in the next version of the paper.